# Effects of Intra-Articular Triamcinolone Injection on Adhesive Capsulitis after Breast Cancer Surgery

**DOI:** 10.3390/diagnostics14141464

**Published:** 2024-07-09

**Authors:** Sungwon Kim, Sunwoo Kim, Jong Geol Do, Ji Hye Hwang

**Affiliations:** 1Department of Physical and Rehabilitation Medicine, Samsung Medical Center, Sungkyunkwan University School of Medicine, Seoul 06351, Republic of Korea; 2Department of Occupational Therapy, Graduate School of Medical Science, Konyang University, Daejeon 35365, Republic of Korea; 3Research Institute for Future Medicine, Samsung Medical Center, Seoul 06351, Republic of Korea

**Keywords:** adhesive capsulitis, breast cancer, intra-articular injection, post-mastectomy, triamcinolone, clinical outcome

## Abstract

Purpose: To investigate the effects of intra-articular glenohumeral joint triamcinolone injection in treating secondary adhesive capsulitis after breast cancer surgery. Methods: This study prospectively enrolled 37 participants, including 22 in the breast cancer surgery group and 15 in the idiopathic group. All participants received intra-articular glenohumeral joint triamcinolone injection in the affected shoulder joint. The clinical outcomes included the Shoulder Pain and Disability Index (SPADI), passive range of motion (PROM), and pain intensity on the Numeric Rating Scale (NRS), which were evaluated before the intervention and 1, 3, and 6 months after. The primary outcome of this study was the mean difference in the total SPADI from baseline to 6 months after the intervention. Results: The mean differences in the total SPADI scores from baseline to 6 months after the intervention were 36.2 ± 16.4 and 47.9 ± 15.2 in the breast cancer surgery group and the idiopathic group, respectively. There was no significant difference between the two groups (*p* = 0.1495). However, the improvements in the SPADI pain subscale at the 3- and 6-month follow-up visits (−31.2 vs. −48.8, *p* = 0.042; −34.1 vs. −50.7, *p* = 0.0006) and the PROM of abduction at the 3-month follow-up (52.4 vs. 70.3, *p* = 0.0072) were inferior in the breast cancer surgery group compared to the idiopathic group. There were no adverse events in either group. Conclusion: Intra-articular triamcinolone injection is an effective and safe treatment option for adhesive capsulitis after breast cancer surgery; however, it has less effect than for idiopathic adhesive capsulitis.

## 1. Introduction

Post-breast cancer surgery patients often experience soft tissue fibrosis, deficits in muscle flexibility, glenohumeral joint stiffness, and adhesive capsulitis of the shoulder [1]. Secondary adhesive capsulitis after breast cancer surgery causes pain and limited motion in the affected shoulder joint, disturbing daily activities and reducing quality of life [2]. Post-breast cancer surgery patients have a greater risk of adhesive capsulitis of the shoulder than the general population. A single-center study showed a 10.3% cumulative prevalence of adhesive capsulitis in breast cancer patients over a 13-to-18-month postoperative period [3]. A cross-sectional observational study of 135 Asian women who underwent breast cancer surgery found a 22.2% prevalence of adhesive capsulitis [4]. Considering the high prevalence and long-term sequelae of secondary adhesive capsulitis after breast cancer surgery, early detection and proper management are essential.

A variety of treatment options is available for adhesive capsulitis, including physical therapy, pharmacologic therapy, intra-articular steroid injection, and surgical management. Idiopathic adhesive capsulitis is caused by the thickening and contraction of the joint capsule due to inflammation and fibrosis; therefore, a combination of intra-articular steroid injections and physical therapy is effective in management [5,6]. However, a consensus on the treatment of secondary adhesive capsulitis after breast cancer surgery has not been established. Intra-articular steroid injections are commonly used to treat idiopathic adhesive capsulitis. However, its effectiveness in patients with secondary adhesive capsulitis after breast cancer surgery remains insufficiently studied. Furthermore, comparisons of the clinical efficacy of intra-articular steroid injection between idiopathic and secondary adhesive capsulitis are lacking.

Patients with adhesive capsulitis after breast cancer surgery have chronic pain and decreased quality of life and require further treatment. The limited evidence regarding the treatment options for this patient population is an obstacle to the implementation of more active physical therapy and injections due to side-effects concerns. Therefore, investigation of the therapeutic effect and safety of intra-articular steroid injection in breast cancer patients may be helpful in the prevention of long-term morbidities and the establishment of a treatment option for these patients.

We hypothesized that intra-articular triamcinolone injection is an effective and safe treatment for adhesive capsulitis after breast cancer surgery. To prove our hypothesis, we evaluated the clinical, functional, and safety outcomes of intra-articular triamcinolone injections in breast cancer surgery patients with adhesive capsulitis. We focused on the therapeutic effects on the pain, range of motion (ROM), and functional disability of the shoulder after breast cancer surgery and compared these effects with those of idiopathic adhesive capsulitis patients.

## 2. Materials and Methods

### 2.1. Study Design and Participants

This study was a single-center, prospective, two-arm clinical trial involving two patient groups with adhesive capsulitis of the shoulder: the breast cancer surgery group and the idiopathic group. Enrollment took place from July 2019 to November 2021 at a tertiary hospital rehabilitation center in South Korea. The inclusion criteria for the breast cancer surgery group were (1) age of 19 years or older; (2) clinical diagnosis of adhesive capsulitis (a significant restriction in passive ROM of the shoulder joint, shoulder pain that was exacerbated by movement, and persisted for at least three months); and (3) affected shoulder joint restriction of at least 30˚ compared to the contralateral side. Participants met this criterion if the joint restriction was present in two or more of the following movements: forward flexion, abduction, or external rotation (with 90° shoulder abduction). The range of motion was measured by a goniometer in the supine position. Additional inclusion criteria were (4) the ability to receive in-hospital physical therapy and (5) surgery for breast cancer. The exclusion criteria were (1) bilateral adhesive capsulitis; (2) secondary adhesive capsulitis caused by trauma (shoulder fracture or dislocation) and/or systemic inflammatory disease (rheumatoid arthritis); (3) other mimicking disorders, such as glenohumeral arthritis, bursitis, rotator cuff disease, or calcific tendinitis of shoulder; (4) inability to perform exercise due to general deconditioning; and (5) communication difficulties. For the idiopathic group, the diagnosis of breast cancer was an exclusion criterion rather than an inclusion criterion. Plain shoulder X-ray and ultrasonography were performed to exclude mimicking disorders. This clinical study was performed in accordance with the principles of the Declaration of Helsinki. Written informed consent was obtained from all participants. The study protocol was approved by the Institutional Review Board of the Samsung Medical Center (approval number: SMC-2019-05-021).

### 2.2. Baseline Characteristics and Clinical Assessments

Participant medical records were reviewed for demographic data, weight, height, calculated body mass index, and medical history. For all participants, the following scores were measured for the assessment of pain and disability of the affected shoulder joint: the Shoulder Pain and Disability Index (SPADI), the passive range of motion (PROM) of the affected shoulder joint, and pain intensity on the Numeric Rating Scale (NRS). The SPADI is a self-reported questionnaire consisting of two subscales with a total of 13 items, including 5 items for pain and 8 items for disability [7]. Participants reported their level of difficulty performing activities of daily living (ADLs) due to pain and limited motion for the previous week. The final score ranges from 0 to 100, with a percentage score of 0 indicating no shoulder disability and 100 indicating complete shoulder dysfunction. As described in a clinical practice guideline by Kelley et al. [8], the PROM of the affected shoulder joint was obtained with an electric goniometer in units of 1° with the participant in a supine position. The shoulder forward elevation (FE), abduction, internal rotation (IR) with 90° shoulder abduction, and external rotation (ER) with 90° shoulder abduction were measured. The pain intensity at rest and during activity was measured by an 11-point NRS for the previous week, with 0 and 10 representing “no pain” and “the worst possible pain”, respectively. The clinical outcomes were assessed at baseline and 1, 3, and 6 months after the glenohumeral joint triamcinolone injection.

The primary outcome of this study was the difference in the total SPADI score from baseline to 6 months after the intervention. The secondary outcomes were as follows: the difference in the total SPADI score from baseline to 1 month and 3 months after the intervention; the difference in the pain subscale of the SPADI, disability subscale of the SPADI, FE of the PROM, abduction of the PROM, IR of the PROM, and ER of the PROM; and pain intensity on the NRS from baseline to 1 month, 3 months, and 6 months after the intervention. Throughout the study period, all side effects possibly related to the intervention were recorded, including worsening pain, skin lesions, lymphedema aggravation, flushing, infection, and hyperglycemia. All clinical parameters were assessed by two blinded, trained occupational therapists during the visits. To ensure consistency and accuracy between the two therapists, a standardized method of PROM measurements was referenced from a clinical practice guideline [8].

### 2.3. Intervention Procedure and Physical Therapy

All participants received physical therapy twice weekly for 12 weeks at our rehabilitation center and were instructed to exercise at home at least once per week. Physical therapy consisted of a warm-up, scapular stabilization, glenohumeral joint stretching, strengthening exercises, and cool-down exercises. Before performing the triamcinolone injection, the complete blood count, erythrocyte sedimentation rate, and C-reactive protein level were measured to exclude infection and reduce the possibility of procedure-associated infection. At the initial visit, all participants received an intra-articular glenohumeral joint injection into the affected shoulder joint under ultrasonographic guidance with 20 mg of triamcinolone and 6 mL of normal saline through a posterior approach. In the posterior approach, the patient lies in a lateral decubitus position facing the physician with the affected side upward. The patient’s ipsilateral arm is internally rotated and adducted. After visualizing the infraspinatus, humeral head, posterior labrum, and joint capsule via ultrasonography, the physician identifies the injection site. Then, the physician puts on a surgical glove and disinfects the area with chlorhexidine-soaked cotton balls using an antiseptic technique. The needle is inserted toward the joint capsule via an in-plane technique. Repeated injections were performed if the participant had any of the following features one month after the initial intervention: a high intensity of pain (>7 on the NRS scale), persistent night or resting pain, or a severe level of self-reported shoulder disability.

### 2.4. Statistical Analysis

Descriptive statistics were used to characterize the demographic and clinical parameters of the participants. Continuous variables were presented as means and standard deviations. To compare the baseline demographic and clinical properties, the Shapiro–Wilk test was used to determine the normal distribution of the continuous variables. The independent *t*-test was used for normally distributed continuous variables, and the Wilcoxon rank sum test was used for non-normal continuous variables. For categorical variables, the chi-square test or Fisher’s exact test was used. To evaluate the effects of the interventions over time in each group, the paired *t*-test and Wilcoxon-signed rank test were used for normal and non-normal values, respectively. To conduct an inter-group analysis of the effects of the interventions over time, generalized estimating equations (GEEs) were performed to confirm statistical differences, and the results were corrected for age, sex, and baseline clinical values. Bonferroni’s correction was used to adjust the p-values of the inter-group and the intra-group analyses. All statistical analyses were performed using SAS version 9.4 (SAS Institute Inc., Cary, NC, USA). Statistical significance was defined by a *p*-value < 0.05.

## 3. Results

### 3.1. Baseline Demographic and Clinical Characteristics

This study prospectively enrolled 37 participants, including 22 in the breast cancer surgery group and 15 in the idiopathic group. The baseline demographic and clinical characteristics are described in Table 1. There was no significant difference in the demographic or clinical properties between the two groups. Of the eight participants who discontinued the study, four were from the breast cancer surgery group and four from the idiopathic group. Five withdrew voluntarily, two due to deteriorating health conditions, and one because of contralateral adhesive capsulitis (Figure 1).

### 3.2. Baseline Primary and Secondary Outcome Measures

The baseline scores of the total SPADI (%) were 44.5 ± 18.3 in the breast cancer surgery group and 50.1 ± 16.9 in the idiopathic group. There was no significant difference in the baseline scores of the total SPADI between the two groups. The baseline scores of the pain subscale of the SPADI (%) were statistically different between the groups, with 44.5 ± 16.9 in the breast cancer surgery group and 50.1 ± 16.9 in the idiopathic group (*p* = 0.042). There was no significant difference in the baseline scores of the disability subscale of the SPADI, FE of the PROM, abduction of the PROM, IR of the PROM, ER of the PROM, the NRS at rest, or the NRS during activity between the two groups (Table 2).

### 3.3. Comparison of Total SPADI Scores and Subscales (Pain and Disability) over Time in Breast Cancer Surgery and Idiopathic Groups

The total SPADI scores at the 1-, 3-, and 6-month follow-up visits were significantly different from the baseline in both groups (*p* < 0.001). The total SPADI scores did not show a significant interaction between time and group (*p* = 0.1495). The mean differences in the total SPADI scores from baseline to 6 months after the intervention were 36.2 ± 16.4 and 47.9 ± 15.2 in the breast cancer surgery group and the idiopathic group. The mean differences in the total SPADI scores from baseline to 1, 3, and 6 months after the intervention did not show significant differences between the two groups. The pain subscale of the SPADI at the 1-, 3-, and 6-month follow-up visits was significantly different from the baseline in both groups (*p* < 0.001). The pain subscale of the SPADI scores showed a significant interaction between time and group (*p* = 0.0009). The mean differences in the pain subscale of the SPADI from baseline to 3 and 6 months after the intervention were significantly different between the two groups (*p* = 0.0042 and *p* = 0.0006). The disability subscale of the SPADI at 1-, 3-, and 6-month follow-up visits was significantly different from the baseline in both groups (*p* < 0.001). The disability subscale of the SPADI scores did not show a significant interaction between time and group (*p* = 0.4328). The mean differences in SPADI disability from baseline to 1, 3, and 6 months after the intervention were not significantly different between the two groups (Figure 2).

### 3.4. Comparison of PROM Measurements over Time in Breast Cancer Surgery and Idiopathic Groups

The PROM measurements at the 1-, 3-, and 6-month follow-up visits showed significant differences from baseline in both groups (*p* < 0.001). Abduction of the PROM showed a significant interaction between time and group (*p* = 0.0058), while FE, IR, and ER of the PROM did not (*p*= 0.7744, *p* = 0.5169, and *p* = 0.1455). The mean differences in the abduction of the PROM from baseline to 3 months after the intervention showed a significant difference between the two groups (*p* = 0.0072). The mean differences in FE, IR, and ER of the PROM from baseline to 1, 3, and 6 months after the intervention did not show a significant difference between the two groups (Figure 3).

### 3.5. Comparison of Pain Intensity as Measured by NRS Scores over Time in Breast Cancer Surgery and Idiopathic Groups

The pain intensity on the NRS at the 1-, 3-, and 6-month follow-up visits was significantly different from the baseline scores in both groups (*p* < 0.05). The NRS at rest and during activity did not show significant interaction between time and group (*p* = 0.3875, *p* = 0.8187). The mean differences in the NRS at rest and during activity from baseline to 1, 3, and 6 months after the intervention did not show significant differences between the two groups (Figure 4).

### 3.6. Number of Injections and Safety Measures

One patient in the breast cancer surgery group and one patient in the idiopathic group received second injections during the follow-up period. There was no significant difference in the proportion of patients who needed second injections between the two groups (*p* = 1.000). There were no adverse events, such as facial flushing, pain flares, skin color changes, joint hematomas, or septic arthritis of the shoulder, after intra-articular injection in either group.

## 4. Discussion

In this study, the clinical outcome comparisons between the idiopathic and breast cancer surgery groups showed similar effectiveness in the SPADI, PROM, and NRS. However, the breast cancer surgery group showed less improvement than the idiopathic group in the SPADI pain subscale and PROM abduction at 3 and 6 months post-intervention. Notably, there were no adverse events observed from the intra-articular steroid injections in this study.

A considerable number of patients suffer from reduced shoulder mobility and restricted upper limb function after breast cancer surgery. In a previous study of breast cancer patients who received axillary lymph node dissection, 57% of patients experienced impaired shoulder mobility, and 69% of patients had impaired shoulder function to perform ADLs three months after the surgery [9]. A systematic review of cases of late morbidity after the treatment of breast cancer reported that there is a significant relationship between restricted ROM and patient-reported functional impairments; the presence of arm problems after breast cancer surgery was related to psychological distress and reduced quality of life [10]. A single-center, cross-sectional, observational study reported adhesive capsulitis in 22.2% of patients after breast cancer surgery [4]. A single-center study in Korea found a cumulative prevalence of 10.3% and a current prevalence of 7.7% of adhesive capsulitis in breast cancer patients with a postoperative period between 13 and 18 months [3]. The rising incidence of breast cancer is a global phenomenon; therefore, the prevalence of secondary adhesive capsulitis in patients with breast cancer will similarly increase and continue to burden society [11]. There are several successful treatment options for idiopathic adhesive capsulitis, such as physical therapy, pharmacologic therapy, intra-articular triamcinolone injections, intra-articular hyaluronate injections, hydrodilatation, and surgical management (including manipulation under anesthesia) [5,12,13]. Among these treatments, intra-articular triamcinolone injection is effective with rapid improvement in pain and shoulder ROM. Intra-articular triamcinolone injections can shorten the natural course of the disease [5,13]. However, the effects of intra-articular triamcinolone injections have not been well studied in patients with adhesive capsulitis after breast cancer surgery. Various physical exercise programs that focus on strengthening shoulder and scapular stabilizers can help to reduce pain and enhance shoulder ROM and function [14,15]. However, early physical therapy with aggressive shoulder stretching and strengthening exercises is limited in patients with adhesive capsulitis after breast cancer surgery due to severe shoulder pain, prominent limited ROM of the shoulder, and the structural changes that occur after breast surgery [16,17]. Our study demonstrates that intra-articular steroid injection for adhesive capsulitis after breast cancer surgery is an effective and safe option to reduce pain and improve the ROM and function of the shoulder joint. In addition, although improvements in the SPADI, PROM, and NRS in the breast cancer surgery group were inferior to some of those of the idiopathic group, they were comparable. Hence, alongside physical therapy, appropriate steroid injections are necessary to enhance the functional outcomes and quality of life for patients who develop adhesive capsulitis after breast cancer surgery.

In our study, the pain subscale of the SPADI at 3 and 6 months and abduction of the PROM at 3 months in the breast cancer surgery group showed inferior results compared to those of the idiopathic group. There are possible explanations for these results. Adhesive capsulitis of the shoulder causes chronic pain, and its pathophysiology after breast cancer surgery is complex. Despite the uncertainty of the pathophysiologic mechanism of adhesive capsulitis, capsular fibrosis and reductions in capsular volume after synovial inflammation are considered to be the primary mechanisms [5]. As a result, non-steroidal anti-inflammatory drugs and corticosteroids are commonly used. Patients with adhesive capsulitis after breast cancer surgery experience greater structural changes than those with idiopathic adhesive capsulitis, including pectoralis tightness, lymphedema, post-mastectomy pain syndrome, and axillary web syndrome [16,18]. After breast surgery, adhesions may occur in the axillary and pectoral areas. Adhesions between the pectoral muscles, subcutaneous tissue, and skin may inhibit the extension of the pectoralis, which limits shoulder flexion and abduction [14,19,20]. Weakness of the scapular stabilizers after mastectomy causes alterations in scapular motion and contributes to adhesive capsulitis [21,22]. These conditions can directly cause pain and limitation of joint motion and may also limit participation in active physical therapy, further aggravating the condition. One study compared the biomechanical properties of the glenohumeral joint capsule during intra-articular corticosteroid hydrodilatation between patients with adhesive capsulitis after breast cancer surgery and patients with idiopathic adhesive capsulitis [23]. The study showed that the capsular pressures at the maximal volume and capsular stiffness were significantly higher in the breast cancer surgery group than in the idiopathic group. Although this prior study did not compare the clinical outcomes between the breast cancer surgery group and the control group, the results of this study suggest that greater structural changes and capsular stiffness could have influenced the worse clinical outcomes depicted in our study.

In this study, we used two different methods to evaluate pain intensity: the pain subscale of the SPADI and the NRS. Our results show a statistically significant difference in the pain subscale of the SPADI but not in the NRS between the two groups. The pain subscale of the SPADI, which consists of five questions modified from the NRS, evaluates pain during shoulder-related activities that reflect real-life functionality. In contrast, the NRS measures general pain intensity at rest and during activity. Therefore, the pain subscale of the SPADI may be more sensitive to treatment effects on shoulder-specific pain than the NRS, particularly in severe cases.

With the limited evidence available, physicians may be reluctant to perform intra-articular steroid injections in patients who underwent breast cancer surgery. Postoperative consequences, including adhesion near the surgical site and ipsilateral upper limb lymphedema, could also magnify the fear of injection-associated adverse events. In our study, the glenohumeral joint injection protocol was carried out using an antiseptic technique, and the injection site was identified before and during the procedure using ultrasonography. To exclude infection and reduce the possibility of procedure-associated joint infection, the complete blood count, erythrocyte sedimentation rate, and C-reactive protein levels were measured before the injection. To reduce potential side effects from the corticosteroid, we used a low-dose injection (20 mg of triamcinolone) in the treatment of adhesive capsulitis of the shoulder. Using this injection protocol, we achieved favorable clinical results without side effects. For improved and consistent outcomes with the procedure, establishing a standardized protocol with the necessary physical examinations and essential laboratory studies before the procedure is crucial.

This study has several limitations. In our study, one patient in each group (the breast cancer surgery group and the idiopathic group) required a second injection due to persistent pain after the initial treatment. Given the small number of patients who needed a second injection, identifying a definitive cause for this initial worsening is challenging. To better understand the factors contributing to initial worsening and develop appropriate treatment strategies, further research involving a larger patient population is necessary. Future studies should explore the potential roles of patient-specific factors, injection techniques, and post-injection rehabilitation protocols in influencing treatment outcomes. This approach will help to identify and address any initial worsening of symptoms and thereby enhance overall patient outcomes. The relatively small sample size also limits the validity of the subgroup analyses. While factors like lymphedema, a history of radiotherapy or chemotherapy, and types of breast cancer surgery are potential subjects for subgroup analyses, they have an unclear impact on the outcomes. Future research with a larger number of participants and more subgroup analyses is needed. In addition, the pharmacological treatments were not controlled. The use of varying doses and types of analgesic medications among the participants could have impacted the outcomes. Third, ultrasound-guided injection is an operator-dependent procedure; therefore, its success may vary across physicians depending on their skill levels. The physicians who participated in this study were experienced physiatrists from the same institution.

## 5. Conclusions

This study shows that intra-articular triamcinolone injection is an effective and safe treatment option for patients who develop adhesive capsulitis after breast cancer surgery. Comparable efficacy was found in clinical outcomes, including the SPADI, PROM, and NRS. However, the breast cancer group exhibited inferior improvement in the pain subscale of the SPADI and abduction of the PROM. Physicians should be aware that intra-articular triamcinolone injection is the treatment of choice in patients with adhesive capsulitis after breast cancer surgery; however, the treatment efficacy may be inferior to that observed in idiopathic adhesive capsulitis.

## Figures and Tables

**Figure 1 diagnostics-14-01464-f001:**
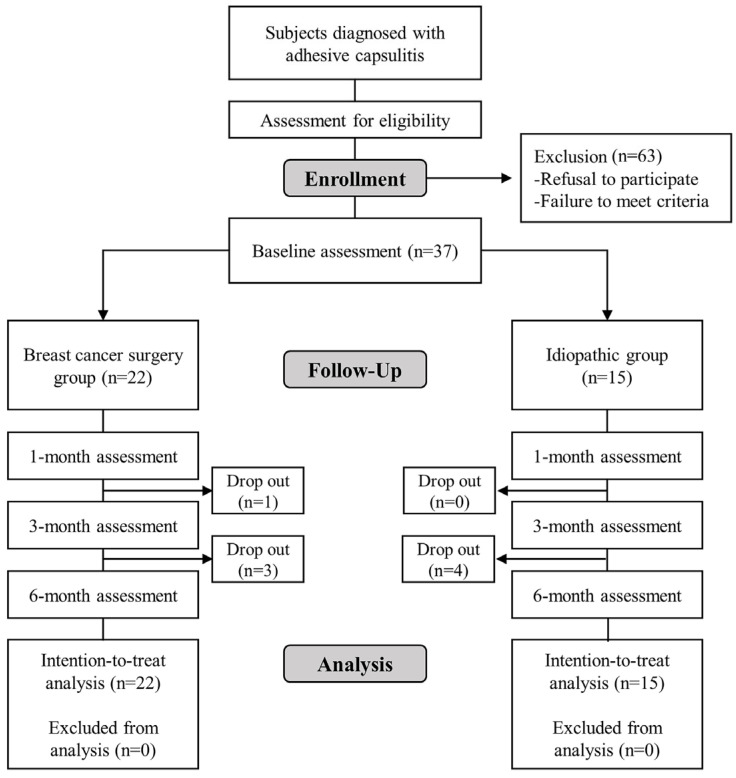
Flowchart of the study.

**Figure 2 diagnostics-14-01464-f002:**
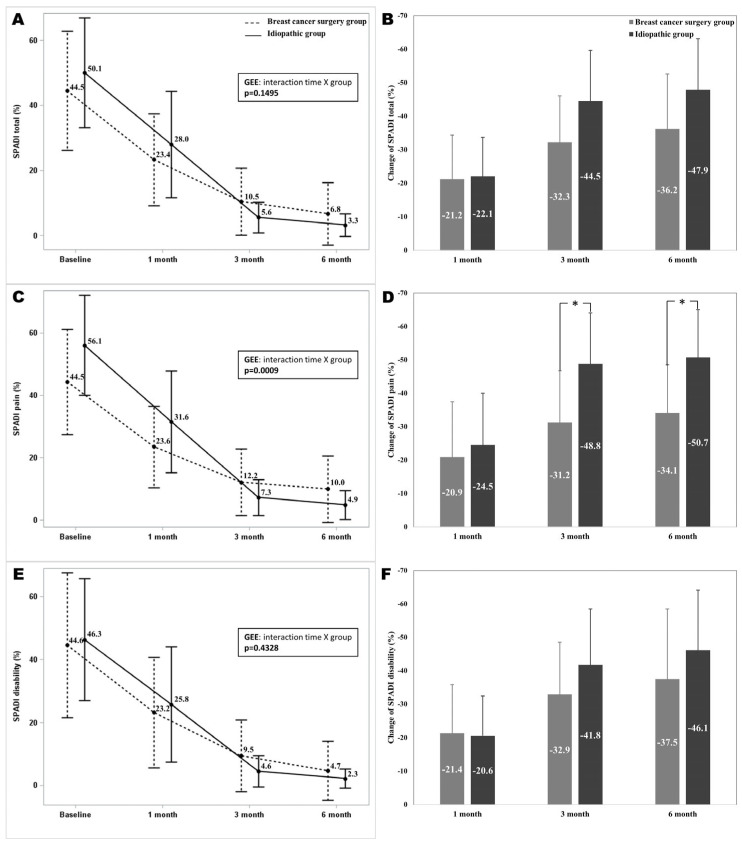
A comparison of total SPADI scores and subscales (pain and disability) over time in both groups. (**A**): The total SPADI scores over time, (**B**): Changes of the total SPADI scores, (**C**): The pain subscale of the SPADI over time, (**D**): Changes of the pain subscale of the SPADI, (**E**): The disability subscale of the SPADI over time, (**F**): Changes of the disability subscale of the SPADI. Figure 2 illustrates significant reduction in total SPADI scores from baseline to 1, 3, and 6 months post-intervention in both breast cancer surgery and idiopathic groups, with no significant interaction between time and group. However, there were significant differences in the SPADI pain subscale between the two groups at 3 and 6 months, indicating a poor improvement in breast cancer surgery group. The error bars represent standard deviation for each group at each time interval. SPADI Shoulder Pain and Disability Index; *: *p* < 0.05.

**Figure 3 diagnostics-14-01464-f003:**
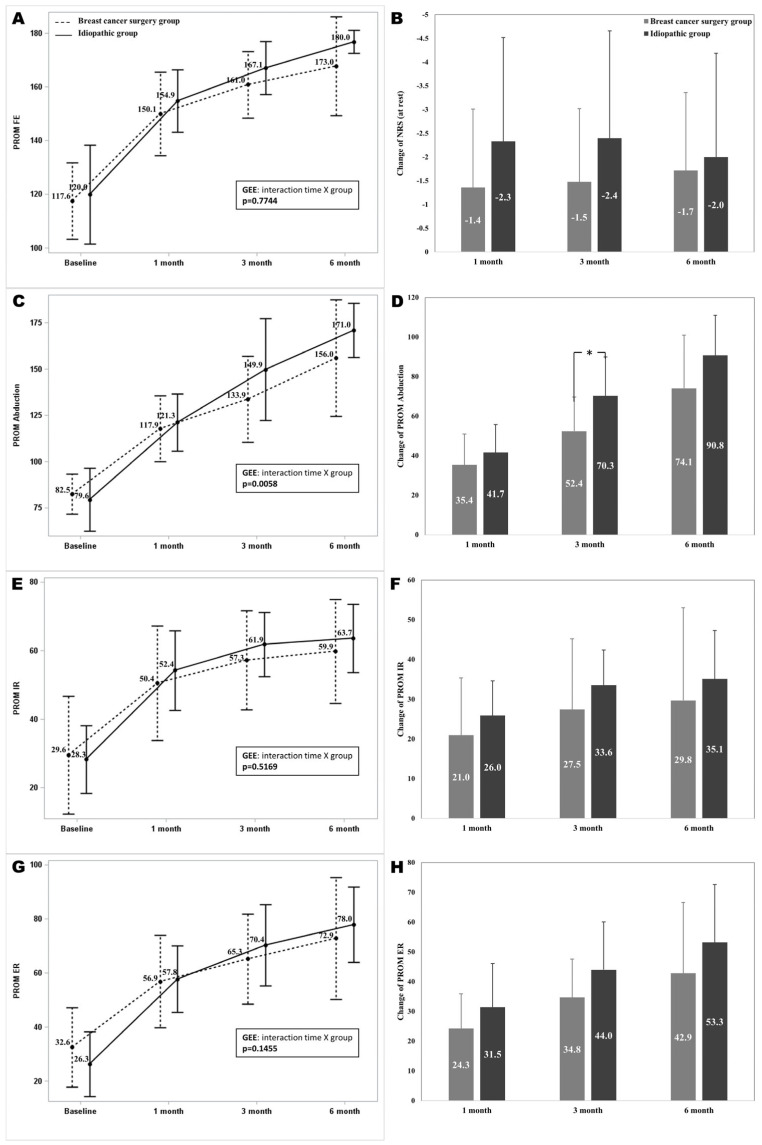
A comparison of PROM measurements over time in both groups. (**A**): Forward elevation of the PROM over time, (**B**): Changes of forward elevation of the PROM, (**C**): Abduction of the PROM over time, (**D**): Changes of abduction of the PROM, (**E**): Internal rotation of the PROM over time, (**F**): Changes of internal rotation of the PROM over time, (**G**): External rotation of the PROM over time, (**H**): Changes of external rotation of the PROM over time. For the PROM of FE, IR, and ER, no significant differences were found between the two groups over the time intervals, showing similar patterns of improvement in PROM post-intervention. A significant interaction between time and group was observed in the abduction of the PROM, indicating differential improvements between the groups 3 months after intervention. The error bars represent standard deviation for each group at each time interval. PROM: passive range of motion; *: *p* < 0.05.

**Figure 4 diagnostics-14-01464-f004:**
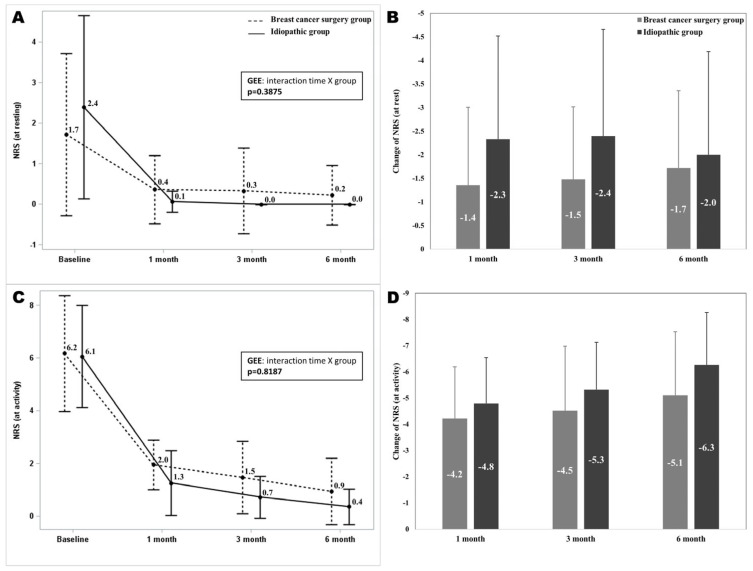
A comparison of NRS scores over time in both groups. (**A**): The NRS at rest over time, (**B**): Changes of the NRS at rest, (**C**): The NRS at activity over time, (**D**): Changes of the NRS at activity. There was no significant interaction between time and group for NRS scores at rest and during activity, indicating similar trends in pain reduction over time for both groups. The error bars represent standard deviation for each group at each time interval. NRS: Numerical Rating Scale.

**Table 1 diagnostics-14-01464-t001:** Baseline demographic and clinical characteristics.

	Total (*n* = 37)	Breast Cancer Surgery (*n* = 22)	Idiopathic (*n* = 15)	*p*-Value
Age (y)	53.5 ± 7.2 (44–69)	52.1 ± 7.3 (44–69)	55.7 ± 6.7 (45–67)	0.088
Female, *n* (%)	34 (91.9%)	22 (100%)	12 (80%)	0.059
Height (cm)	160.3 ± 7.3	160.1 ± 6.5	160.7 ± 8.5	0.588
Weight (kg)	58.4 ± 11.5	56.6 ± 9.1	60.9 ± 14.2	0.676
BMI (kg/m^2^)	22.6 ± 3.5	22.1 ± 3.4	23.4 ± 3.6	0.284
Affected side dominant, *n* (%)	21 (56.8%)	13 (59.1%)	8 (53.3%)	0.729
Affected side, right, *n* (%)	19 (51.4%)	11 (50.0%)	8 (53.3%)	0.842
Dominant hand, right, *n* (%)	35 (94.6%)	20 (90.9%)	15 (100%)	0.505
Diabetes, *n* (%)	4 (10.8%)	4 (18.2%)	0 (0%)	0.131

Values are presented as mean ± standard deviation (range) or *n* (%). BMI: body mass index.

**Table 2 diagnostics-14-01464-t002:** Baseline primary and secondary outcomes.

Parameter	Total (*n* = 37)	Breast Cancer Surgery (*n* = 22)	Idiopathic (*n* = 15)	*p*-Value
SPADI, total (%)	46.8 ± 17.7	44.5 ± 18.3	50.1 ± 16.9	0.355
SPADI, pain (%)	49.2 ± 17.3	44.5 ± 16.9	56.1 ± 16.1	0.042 *
SPADI, disability (%)	45.3 ± 21.3	44.6 ± 23.0	46.3 ± 19.4	0.812
PROM				
FE, deg	118.6 ± 15.9	117.6 ± 14.3	120.0 ± 18.4	0.663
Abduction, deg	81.3 ± 13.5	82.5 ± 10.8	79.6 ±17.1	0.524
IR, deg	29.1 ± 14.6	29.6 ± 17.3	28.3 ± 9.9	0.578
ER, deg	30.0 ± 13.9	32.6 ± 14.7	26.3 ± 12.0	0.184
NRS (at resting)	2.0 ± 2.1	1.7 ± 2.0	2.4 ± 2.3	0.345
NRS (at activity)	6.1 ± 2.1	6.2 ± 2.2	6.1 ± 1.9	0.871

Values are presented as mean ± standard deviation. SPADI: Shoulder Pain and Disability Index; PROM: passive range of motion; FE: forward flexion; IR: internal rotation; ER: external rotation; NRS: Numerical Rating Scale; *: *p* < 0.05.

## Data Availability

The datasets generated and analyzed during this study are not publicly available due to privacy protection and medical confidentiality but are available from the corresponding author for reasonable requests and with patient permission.

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
