# Peer review of "Effects of Intra-Articular Triamcinolone Injection on Adhesive Capsulitis after Breast Cancer Surgery"

_diagnostics, 2024, doi:10.3390/diagnostics14141464_

Round 1
Reviewer 1 Report
Comments and Suggestions for Authors
1. Second author Sunwoo Kim has been labeled with affiliation 2 and 3. What is the name of 3?
2. Are the PROM measurements for all patients performed by one person with standard procedure?
3. Line 168 of the manuscript described that “ Only one participant in the idiopathic group was male.” However, on Table 1 revealed that 12/15 is female in the idiopathic group, in other words, there should have 3 males in the idiopathic group. Which one is correct? Since there are no male patients in the breast cancer surgery group, why not limit female patients in the idiopathic group? Furthermore, did you really put variable “sex” in the GEE modeling procedures (as mentioned in Section 2.4. Statistical analysis)? In that case, the effect estimators obtained from the GEEs could be unstable and imprecise due to so small number of male patients.
4. Is time series analysis appropriate for this study? Especially if you want to see whether a trend of treatment effect over time exists or not.
5. For those who needed second injection, what have caused initial worsening? How to avoid possible initial worsening if you want to apply this treatment to other patients in the future? This issue should be discussed.
6. English should be edited.
Comments on the Quality of English LanguageError words and grammar are to be corrected.
Reviewer 2 Report
Comments and Suggestions for Authors
GENERAL COMMENTS
As the authors mentioned, adhesive capsulitis occurring after surgery in breast cancer patients is a common complication. However, only a few studies have been reported. In addition, a previous study reported that compared to idiopathic adhesive capsulitis, secondary adhesive capsulitis arising from breast cancer had greater capsule stiffness and delayed response to treatment and surgery such as mastectomy, known to be one of the major risk factors for secondary adhesive capsulitis. Therefore, the study in which the authors evaluated the effect of triamcinolone injection in patients with adhesive capsulitis after breast cancer surgery is meaningful. However, there are some limitations that need to be revised.
SPECIFIC AREAS TO BE REVISED
Title
: If the authors intend to report the effect of intraarticular triamcinolone injection on adhesive capsulitis occurring after breast cancer surgery, it would be better to delete the term ‘idiopathic adhesive capsulitis.’
: ‘ Effect of intra-articular triamcinolone injection on adhesive capsulitis after breast cancer surgery’
Abstract
Purpose
: Ok.
Methods
‘All participants received intra-articular glenohumeral joint injection in the affected shoulder joint and received physiotherapy for 12 weeks’
: Please add that triamcinolone was injected.
: Also, please specify what clinical evaluations were assessed.
Results
: Please add passive ROM range, SPADI values ​​and p value.
Conclusion
: Too verbose. Please describe more concisely.
Keywords
: Add ‘Triamcinolone’.
: Please add ‘clinical outcome’ instead of ‘pain’.
Introduction
1st paragraph
: There is too much unnecessary content. Also, delete any content that overlaps with the second paragraph.
: Please describe the contents related to adhesive capsulitis that occurs after breast cancer surgery.
Line 50-52, ‘Despite the high prevalence of adhesive capsulitis of the shoulder after breast cancer surgery, the risk factors for the condition are largely unknown. Presence of lymphedema, age 50-59 years, and mastectomy are possible risk factors for secondary adhesive capsulitis after breast cancer surgery [8-10].’
: Since this study did not evaluate the risk factors of secondary adhesive capsulitis after breast cancer surgery, please delete the above sentence.
Line 64-65, ‘Furthermore, comparisons of the clinical efficacy of intra-articular steroid injection between idiopathic and secondary adhesive capsulitis are lacking.’
: There is a paper titled ‘Effects of Hydrodilatation With Corticosteroid Injection and Biomechanical Properties in Patients With Adhesive Capsulitis After Breast Cancer Surgery.’ published in Ann Rehabil Med in 2022. Please review the content of the above study and describe the differences from the authors' study.
4th paragraph
: As the authors mentioned, evaluating the safety of steroid injection after breast cancer surgery is very important. Please add results related to side effects to the abstract.
Methods and materials
Study design and participant
(1) ‘Age of 19 years or older’
: Is there a reason for setting the age range as such?
(2) Clinical diagnosis of adhesive capsulitis
: Please describe the clinical diagnosis in more detail.
(3) affected shoulder joint restriction of at least 30Ëš compared to the contralateral side
: Is there a reference? Also, didn’t the authors include internal rotation to evaluate shoulder joint restriction?
Baseline characteristics and clinical assessments
: Please add contents related to PROM, NRS, and side effects to the abstract along with the results.
: There is a study reporting differences in clinical outcomes after treatment in adhesive capsulitis that occurred after breast conserving surgery and mastectomy. Did the authors distinguish between breast conserving surgery and mastectomy among the breast cancer surgeries? .
: Did the authors perform a reliability analysis between physiotherapists and occupational therapists that evaluated the clinical parameters?
Results
Baseline characteristics and clinical assessments
: For the demographic data in Lines 167-178, it is sufficient to describe it in a table.
: Also, didn't the authors evaluate metabolic diseases such as hyperlipidemia other than diabetes mellitus among comorbidities?
: Please include the number of patients meeting the exclusion criteria in the figure.
Baseline primary and secondary outcome measures
: While SPADI(pain) was significantly lower in the breast cancer surgery group, there was no statistically significant difference in NRS. What do the authors think is the reason?
Number of injections and safety measures
‘One patient in the breast cancer surgery group and one patient in the idiopathic group received second injections during the follow-up period, and there were no significant differences in the proportion of patients who needed second injections between the both groups.’
: Please provide the p-value.
Discussion
‘This study revealed that intra-articular triamcinolone injection significantly improved pain, shoulder function, and PROM with adhesive capsulitis, regardless of the history of breast cancer surgery.’
‘Thus, for breast cancer patients, continuous pain and ROM management with close monitoring is essential beyond 3 months post-injection.’
: Please delete unnecessary sentences.
Line 289-294, ‘The evaluation and treatment for adhesive capsulitis after breast cancer surgery found to be insufficient. Various physical exercise programs that focus on strengthening shoulder and scapular stabilizers can help reduce pain and en-hance shoulder ROM and function. However, early physical therapy with aggressive shoulder stretching and strengthening exercise are limited in patients with adhesive capsulitis after breast cancer surgery due to severe shoulder pain, prominent limited ROM of the shoulder, and structural change after breast surgery.
: Please add reference to each sentence.
Line 297-298, ‘Hence, alongside physical therapy, appropriate interventions are necessary to enhance functional outcomes and quality of life for adhesive capsulitis after breast cancer surgery.
: Did the authors perform additional interventions other than steroid injection in this study?
Line 320-321, ‘A previous study evaluated the effects of hydrodilatation in patients with adhesive capsulitis after breast cancer surgery, but the study did not compare the effects with a control group.’
: The above study also established a control group. Please describe in more detail the differences between the above study and the authors' study.
Line 337-345, ‘Previously, there have been some efforts to apply physical therapy and exercise in patients who underwent breast cancer surgery. Physical therapy including passive mobilization, stretching, and exercise programs introduced as early as 7 days after surgery can be effective for improvement of shoulder ROM and postoperative pain [25, 26]. A randomized controlled trial showed that addition of scapular stabilization exercises and Thera-band strengthening exercises can be beneficial for shoulder function and quality of life in post-mastectomy patients with adhesive capsulitis [16]. Proper pharmacologic intervention for inflammation and the prescription of proper rehabilitation programs are important to improve ROM in patients with adhesive capsulitis after breast cancer surgery, even after 3 months post-treatment.’
: This study reported the results after steroid injection, not the results after physical therapy and exercise. Please delete any content related to proper pharmacologic intervention and rehabilitation as it is not related to this study.
: If the authors want to mention the importance of a standardized protocol during steroid injection, related content must first be described in the method session. Please add a standardized protocol related to steroid injection to the method session.
Limitation
: Didn't the authors perform power analysis or sample size analysis?
Line 366-367, ‘This study strength is its six-month follow-up duration, enabling the assessment of long-term effects of intra-articular steroid injections in breast cancer surgery patients.’
: A follow up duration of 6 months is not long term. Please delete it.
Conclusion
: As mentioned above, too verbose.
Line 378, 380, ‘Notably, the breast cancer group showed a poor improvement in the SPADI pain subscale and PROM abduction at 3 and 6 months post-intervention, indicating that continuous monitoring and follow up are necessary even after this period.’
: Rather than the necessity of continuous monitoring, isn't there less of an effect after intraarticular steroid injection in patients with adhesive capsulitis after breast surgery compared to idiopathic adhesive capsulitis?
Comments on the Quality of English LanguageModerate editing of English language required
Reviewer 3 Report
Comments and Suggestions for Authors
Thank you for your effort on the manuscript.
My comments:
Introduction section:
This section is well written. The first paragraph may be removed to shorten this section but this is not a must, I am just suggesting it to authors. The decision is up to them.
The materials and methods section:
This section is well-written.
The results section:
In table one female rate for idiopathic group reported as 80%(which indicates 3 male patients) but in text you stated there was only one male patient in idiopathic group. Please check it. And exclusion of male patient from study may be better.
Why did not you exclude the patients who discontinued to study?
Was there any difference the patient who recieved once and twice physiotherapy in a week?
Discussion section:
Please improve this section more discussing your results with literature. There is nearly no discussion in this section.
Round 2
Reviewer 2 Report
Comments and Suggestions for Authors
Thank you for revising the manuscript according to the reviewer's comments.
Comments on the Quality of English LanguageMinor editing of English language required
Reviewer 3 Report
Comments and Suggestions for Authors
Thank you for your effort on the manuscript and the revision.
I think current form can be accepted to pnulication.